# Unconscious Bias among Health Professionals: A Scoping Review

**DOI:** 10.3390/ijerph20166569

**Published:** 2023-08-12

**Authors:** Ursula Meidert, Godela Dönnges, Thomas Bucher, Frank Wieber, Andreas Gerber-Grote

**Affiliations:** School of Health Sciences, Zurich University of Applied Sciences, Katharina-Sulzer-Platz 9, 8400 Winterthur, Switzerland; godela.doennges@zhaw.ch (G.D.); thomas.bucher@zhaw.ch (T.B.); frank.wieber@zhaw.ch (F.W.); andreas.gerber-grote@zhaw.ch (A.G.-G.)

**Keywords:** unconscious bias, implicit bias, health professionals, health disparities

## Abstract

Background: Unconscious biases are one of the causes of health disparities. Health professionals have prejudices against patients due to their race, gender, or other factors without their conscious knowledge. This review aimed to provide an overview of research on unconscious bias among health professionals and to investigate the biases that exist in different regions of the world, the health professions that are considered, and the research gaps that still exist. Methods: We conducted a scoping review by systematically searching PubMed/MEDLINE, CINAHL, PsycINFO, PsycARTICLES, and AMED. All records were double-screened and included if they were published between 2011 and 2021. Results: A total of 5186 records were found. After removing duplicates (*n* = 300), screening titles and abstracts (*n* = 4210), and full-text screening (*n* = 695), 87 articles from 81 studies remained. Studies originated from North America (*n* = 60), Europe (*n* = 13), and the rest of the world (*n* = 6), and two studies were of global scope. Racial bias was investigated most frequently (*n* = 46), followed by gender bias (*n* = 11), weight bias (*n* = 10), socio-economic status bias (*n* = 9), and mental illness bias (*n* = 7). Most of the studies were conducted by physicians (*n* = 51) and nurses (*n* = 20). Other health care professionals were rarely included in these studies. Conclusions: Most studies show that health professionals have an implicit bias. Racial biases among physicians and nurses in the USA are well confirmed. Research is missing on other biases from other regions and other health professions.

## 1. Introduction

Although the professional ethos of health care providers includes treating all people equally regardless of their physical and mental characteristics, there are inequalities between groups of people in health care [1,2]. There are numerous factors that explain health disparities between groups of people, such as poverty, low health literacy, and harmful health behaviors [3]. However, disparities remain when confounding variables related to patient access, such as income and health insurance, are accounted for [3,4]. One factor contributing to these disparities may be explained by unconscious biases in provider perceptions [5,6,7].

The term “bias” is used to describe a tendency to favor one group over another [8]. It often involves associating physical features or characteristics with a particular behavior [2,9,10], i.e., categorization. The ability to quickly categorize a person or phenomenon that we encounter is likely an evolutionary development that ensured survival [11]. Such categorizations are effectively a shortcut in the brain and, therefore, they relieve the individual of the burden of making repetitive decisions [8,12]. Related terms in the literature include stereotype and prejudice [9]. Stereotypes are often negative and overgeneralized views held toward individuals of certain disadvantaged groups, e.g., poor people are lazy [8,13,14]. Prejudices are negative attitudes toward another person or group formed in-advance of any experience, and they comprise three components: affective (e.g., approach/avoid predilection), cognitive (e.g., assumptions and beliefs), and behavioral (e.g., propensity) [15]. 

The literature distinguishes between explicit and implicit biases. An explicit bias is based on the conscious thoughts and beliefs that an individual can report [16]. Implicit or unconscious bias, on the other hand, is not conscious of an individual. Often, implicit bias is described as involuntary associations or attitudes that influence our perceptions, and thus our behaviors, decisions, and interactions in an unconscious manner (e.g., [1,8,10,12,13]). It is believed that implicit biases are learned through cultural immersion and socialization [10]. These unconscious associations or attitudes are expressed, for example, in nonverbal behaviors toward others, such as eye contact or physical proximity [10].

A person’s explicit and implicit biases can be different or even contradictory to each other. For example, negative views and beliefs associated with disadvantaged groups may be consciously rejected and inevident in self-reported questionnaires [16], but may still exhibit response–time patterns [10,17] on implicit measures, such as the Implicit Association Test (IAT) [18] or the Go/No-Go Association Task (GNAT) [19]. For example, Green et al. showed that physicians who did not explicitly report bias against African American patients and did not classify them as less cooperative still exhibited an implicit belief that these patients were less cooperative compared to white patients [7].

Even when health professionals are aware of their biases and consciously try to suppress them [17], high cognitive loads, such as fatigue, heavy workload, distraction, or time pressure, can still activate implicit biases [2,9,16,20]. This is because cognitive load makes it more difficult to process social information thoroughly and increases the susceptibility to responses based on learned categorizations [3,16]. Bias, then, can unconsciously influence the way information about a person is processed, leading to unintended inequalities. The consequences of these unconscious biases can affect medical school admissions, patient care, faculty hiring, promotion, and advancement opportunities [8]. In health-sciences education, biases and stereotypes may shape education when students learn with typical examples based on prevalence data [8]. These prevalences can lead to premature conclusions and missed diagnoses when applied to heterogeneous patients who are not perceived as individuals, but as demographic characteristics [8]. 

For health professionals, inequalities that work at the disadvantage of those who are already at-risk are particularly significant. Examples include ethnic minorities, immigrants, the impoverished, those with low health literacy, sexual minorities, children, women, the elderly, the mentally ill, the obese, and the disabled [10]. Unconscious bias can influence providers’ decisions about treatment, screening, and other procedures they recommend to these patients [3]. 

Over the last two decades, much research has been conducted on unconscious bias. However, an up-to-date overview is missing. Therefore, this research aims to provide a current overview of the research on unconscious bias and to identify the remaining knowledge gaps. In a prior literature review, Fitzgerald and Hurst [10] examined whether trained health care professionals demonstrate implicit/unconscious biases toward certain types of patients and the unconscious biases they exhibit toward patients. As they only included data through March 2013, we are interested in how the research field has evolved and which biases have been examined. In addition, because most of the research has been conducted in the United States, we are interested in how research on this topic has developed in Europe. Thus, our question is as follows: What biases are prevalent in which countries? Moreover, a review by Fitzgerald and Hurst [10] showed that previous research focused on two health professions: physicians and nurses. In this review, we examine whether other health professions have been studied in the last 10 years. So, our question is as follows: Which health professions are included in these studies? Finally, we want to identify the substantive research gaps that remain in this area.

## 2. Materials and Methods

We chose the method of a scoping review [21] as our aim was to create an overview of the research conducted on unconscious bias during the last 10 years, to include a broad range of studies, to evaluate the scope of available research, and to identify research gaps [22]. In doing so, we followed the JBI International Scientific Committee’s guide on conducting scoping reviews that builds on the PRISMA-ScR guideline [22] and published a research protocol on OFS before performing the search [23].

To operationalize our research questions, we used the Population, Construct, and Context (PCC) mnemonic for our search components. As the population, we selected trained health professionals such as physicians, nurses, midwives, occupational therapists, and physiotherapists (see Appendix A Table A1). The construct was unconscious or implicit bias. For context, we focused on the health care delivery system. From these components, we derived the following three inclusion and exclusion criteria: 1. Studies were included in which participants were trained health professionals. Studies with untrained health care workers, students, or administrative staff were excluded. Studies with mixed populations were included if 2/3 of participants were trained health professionals or if outcomes were presented separately rather than combined. 2. For the construct, we included empirical studies that used methods that elicit unconscious bias towards patients. Studies investigating explicit bias, structural discrimination, and other forms of overt discrimination by health care workers against patients were excluded. Regarding the measurement, we included studies that used implicit measures [24] such as the “Implicit Association Test” (IAT) [18] or the “Go/No-Go Association Task” (GNAT) [19] and other implicit measures [25]. In addition, we included vignette studies. Vignettes are short descriptions of a person or situation, which can be used to elicit subjects’ attitudes [26,27]. 3. The setting is the health care system. Studies conducted in home or community settings were excluded. Furthermore, we included studies published between 1 January 2011 and 31 December 2021 in English, German, or French.

A sensitive search approach was used. We selected the databases PubMed/MEDLINE, CINAHL, PsycINFO/PsycARTICLES, and AMED. Following the PCC-mnemonic (Population, Construct, and Context). For the population, this was “health professionals” and specific health professions (e.g., “nurse”, “therapist”, and “physician”); for the construct, this was “unconscious bias”, “implicit bias”, and related terms (e.g., “stereotype”, “prejudice”, and “racism“); and for the context we used “health care delivery” or similar terms (e.g., “patient care”). To these search terms we added relevant subject headings and MeSH terms from the relevant databases and combined them with the Boolean operators “AND” and “OR”. For detailed search strings see Appendix A. We then tested the search string, and it was additionally checked by a second researcher using the Peer Review of Electronic Search Strategies (PRESS) checklist [28]. The search was conducted on the 14 January 2022 through to the 18 January 2022.

All identified records were imported into the literature management program Zotero [29] and from there into the web application “Covidence” [30] (see Figure 1). After the removal of duplicate records, we conducted a title and abstract screening, in which each record was double screened by two of the three independent reviewers from our team (U.M., C.D., and T.B.) using the established inclusion and exclusion criteria. 

Disagreements regarding inclusion or exclusion were resolved through discussion until an agreement was reached. In a second step, the full texts were screened by two independent reviewers following the same procedure. They also documented the reasons for exclusion. With review articles we proceeded as follows: we extracted all studies that met our inclusion and exclusion criteria and included them in our analysis. Data were extracted and summarized using a structured extraction table.

## 3. Results

### 3.1. Search Results

The database search resulted in 5186 records (see Figure 1). After the removal of 300 duplicates, 4210 records were removed during the title and abstract screening, resulting in 676 records. From the reviews, 19 additional studies were extracted at this point of the review process, leading to 695 records. Of these 695 records, 608 records were excluded for the following reasons: 549 studies predominantly used self-report questionnaires with explicit measures, 26 records addressed the wrong population, measuring patients’ bias either in students of health professions or untrained health professionals, 24 records were excluded as they were not empirical studies, 6 records were written in another language than English, French or German, 2 studies used an implicit measure (IAT) but did not publish the results, and 1 study was conducted in a community setting. As a result, 87 articles that were based on 81 studies were included. For six studies, more than one article was published reporting different aspects but using the same data.

### 3.2. Study Characteristics

Of the 87 records included in this review, 86 were peer-reviewed and published as articles; only one record [31] was a dissertation published as a monograph. Of the 81 studies that are covered in these records, most studies were cross-sectional (*n* = 68). A few studies were re-analyses of previously collected data (*n* = 5), used repeated measures of unconscious bias (*n* = 5), had a prospective design or longitudinal design (*n* = 1 and *n* = 1, respectively), or were a cohort study (*n* = 1). The sample sizes of the studies ranged from 11 to 25,006 participants with a total of 58,908 participants and a mean of 727 participants. 

Regarding the measures, 55 of the 81 included studies used an implicit measure: 53 used one or several versions of the IAT [18], 1 study [32] the GNAT [19], 1 study [33] the Affect Misattribution Procedure (AMP) [34], and 1 study [35] used a reaction time task with subliminal priming. Vignettes in which at least one relevant variable (e.g., gender, Body Mass Index) was changed and treatment recommendations were assessed were used in 46 studies. Of these, 25 studies used vignettes only, and 21 studies combined an IAT and vignettes so that the unconscious bias could be analyzed in combination with the individual treatment recommendations using the vignette method.

### 3.3. Results of the Studies

Table 1 provides information on all included studies and their findings related to unconscious bias. In the 81 studies included, 70 studies found some type of unconscious bias against minorities, while 11 studies found no bias. Regarding the 56 studies that used an IAT or another implicit measure, 54 studies found at least one unconscious bias, while 2 found no bias at all. Looking at the level of individual implicit test measures, 89 tests were conducted in total. Thereof, 85 found an unconscious bias and 2 did not find any unconscious bias; from 2 further studies, the results were not applicable as they were not published. Of the 84 tests showing an unconscious bias, 16 weak, 35 moderate, and 22 strong unconscious biases were found. For 14 tests, the strength of the unconscious bias was not applicable as the results did not follow the recommended form or a classification in terms of strength was not provided for the instruments. 

Regarding the 45 studies that used one or several vignettes, 38 studies found at least one unconscious bias in terms of a difference in treatment recommendations or treatment decisions according to the bias assessed, whereas 21 did not find any differences in treatment recommendations or decisions. 

Regarding the 20 studies that used both an IAT and a vignette, 13 studies found a bias in the IAT and some difference in the vignettes according to the bias assessed, whereas 7 studies found a bias in the IAT but no differences in the vignettes. One study found some differences in the vignettes and in the IAT for one bias (race) but not for another (SES) [36]. One study [37] used treatment outcomes from medical records prior to assessing unconscious racial bias with the IAT, finding no difference in treatment between African American, Hispanic, and Caucasian patients treated for hypertension depending on the level of implicit bias. 

There were six studies that tested interventions to reduce or adjust unconscious bias. Each of these interventions are briefly detailed further below. In study [38], participants engaged in an empathy-inducing perspective-taking intervention that instructed them to imagine how patients’ pain affected patients’ lives, which resulted in a 55% reduction in pain treatment bias in comparison to controls. Study [39] found a similar effect in a subset analysis of white participants who demonstrated a significantly decreased level of implicit bias against Latino people in terms of pleasantness in the intervention group compared to the control group when looking at and reflecting on a photographic story of a migrant family. In study [40], an educational module on patient-centered counseling served as an intervention, resulting in the correlation between personal preferences for pre-natal testing and patient-recommended testing being reduced to non-significance. Thus, the relationship between implicit bias and the propensity to recommend genetic testing was reduced. In study [41], one group had the possibility to exchange real-time information in structured peer networks, which resulted in significantly improved clinical accuracy and showed no bias compared to the control group, which showed significant disparities in treatment recommendations. Two studies were identified that were not successful in reducing unconscious bias. Specifically, study [42] involved clinical residents exposed to schizophrenic patients as an intervention, which resulted in an increased implicit association of illness with the word “criminal”. Finally, study [43], focused on the uncontrollable aspects of obesity, found no change in weight-related IAT as a result of the intervention. 

**Table 1 ijerph-20-06569-t001:** Overview of included studies, with information on country, type of bias, method, sample, results, and effect size.

Reference	Country	Type of Bias	Method	Description	N	HPs	Results	+/++/+++/0
Afulani et al., 2021 [44]	Kenya	SES bias	IAT and vignettes	Cross-sectional mixed-methods study with SES-IAT, vignettes varying by SES and follow-up qualitative interviews	101	Nurses, midwives, physicians, support staff	Moderate negative UB against women with low SES in IAT, differences in expectations towards patients according to stereotypes for both low- and high-SES women in vignettes	++/n.a.
Al Alwan et al., 2019 [45]	Saudi Arabia	SES bias	Vignettes	Cross-sectional vignette study varying by SES (low/high/neutral)	45	Physicians	No difference in accuracy or time for diagnosis between low, high, or neutral SES	0
Anastas et al., 2020 [46]	USA	Racial bias, SES bias	IAT and vignettes	Cross-sectional study with race IAT (black/white) and re-analysis of computer-simulated vignette study varying by race (black/white) and SES (high/low)	436	Physicians	Weak negative UB for black people and strong negative UB for low SES in IATs. Effects between race and SES IAT scores and pain judgement. Different treatment decisions for race and SES in vignettes	+/+++/n.a.
Aweidah et al., 2016 [47]	Australia	Weight bias	IAT	Cross-sectional mixed-methods pilot-study with weight IAT and qualitative interviews	37	Diagnostic radiographers	Negative weight bias	n.a.
Barnato et al., 2011 [48]	USA	Racial bias	Vignettes	Cross-sectional study with randomized trial with standard ized patients varying by race (black/white)	33	Physicians	No difference in treatment decisions	0
Bartley et al., 2015 [49]	USA	Ageism, gender bias, racial bias	Vignettes	Cross-sectional online vignette study with videos of virtual humans varying by age, gender, and race	154	Dentists, physicians	Differences in ratings of pain and pain treatment by age, sex and race of patient and by provider age, sex and race	n.a.
Bernardes & Lima 2011 [50]	Portugal	Gender bias	Vignettes	Cross-sectional vignette study varying by gender	126	Nurses	Differences in attribution of pain according to gender in the absence of diagnosis	n.a.
Blair et al., 2013, 2013 and 2014 [37,51,52]	USA	Racial bias	IAT	Cross-sectional mixed-methods study with race IATs (black/white and Hispanic/white), patients’ perception of treatment and electronic medical records	210	Health care providers	Strong negative UB against black and Hispanic people. The stronger the physicians’ UB, the lower the rating of the black patient of treatment (not for Hispanic). No difference in treatment according to medical records	+++/+++/0
Bøker Lund et al., 2018 [53]	Denmark	Weight bias	IAT and vignettes	Cross-sectional study with IAT on attitude and stereotypes for obesity and vignettes varying by gender and weight	240	Physicians	Strong negative UB against over weight people in IAT on attitudes and stereo-types. No differences in treatment options by weight, but in recommen dations by sex and weight in vignettes	+++/0/n.a.
Bous et al., 2021 [54]	USA	Disability bias	IAT	Cross-sectional online study with disability IAT (cleft lips/normal lips)	52	Dentists	Moderate negative UB against people with disability (cleft lips)	++
Breathett et al., 2019 [55]	USA	Racial bias	Vignettes	Cross-sectional mixed-methods study with vignette varied by race (black/white) and think-aloud interviews	422/44	Health care providers	No differences in recommendations for treatment in vignettes according to race	0
Brener et al., 2013 [56]	Australia	Mental illness bias	IAT	Cross-sectional study with IAT on mental illness	74	Mental health providers	Weak negative UB bias against people with mental illness	+
Burgess et al., 2014 [20]	USA	Racial bias	Vignettes	Cross-sectional web-based randomized vignette study varying by race (black/white) and cognitive load	99	Physicians	Differences in drug prescription according to physicians’ gender, cognitive load, and patient race	n.a.
Cassell 2015 [31]	USA	Racial bias	Vignettes and IAT	Cross-sectional online-study with race IAT (black/white) and vignettes varying by race	216	Physicians	Moderate negative UB on IAT against black people. Difference in diagnosis but no difference in treat ment recommendation in vignettes	++/n.a./0
Centola et al., 2021 [41]	USA	Racial bias, gender bias	Vignettes	Cross-sectional online-study with video-based vignettes manipulated by gender and race (black/white)	120	Physicians	Negative UB against black women with unsafe treatment recommen dations compared to white male patient	n.a.
Chapman et al., 2018 [33]	USA	Racial bias	AMP	Sequential cohort online study with race AMP (Hispanic/ white) with pre- and post-intervention measure	69	Physicians	Negative racial bias against Hispanic people	n.a.
Claréus & Renström 2019 [57]	Sweden	Gender bias	Vignettes	Cross-sectional online study with vignette varying by gender	90	Physicians	Negative UB on women in diagnosis related to back pain	n.a.
Colón-Emeric et al., 2017 [58]	USA	Racial bias, gender bias	Vignettes	Repeated-measures study (pre- and post-intervention) with randomized vignettes differing by race, gender, and age	541	Nurses, rehabilitation staff	Small degree of negative UB against black people, no UB on gender	+
Cooper et al., 2012 [39]	USA	Racial bias	IAT	Cross-sectional study with race IAT (black/white) on attitudes and racial stereotyping on compliance	40	Physicians	Moderate negative UB on race attitude and race compliance stereotyping	++/++
Crapanzano et al., 2018 [59]	USA	Mental illness bias	IAT	Cross-sectional online-study with 4 IATs on attitude (good/bad), permanence, controllability, and ethology of mental illness (depression/ physical illness)	86	Physicians, psychiatrists	Weak–moderate negative UB on attitude, permanence, and controllability against mentally ill people amongst physicians not in psychiatrists. Moderate negative UB on ethology for both	+-++/+-++/+-++/0/++
Daugherty et al., 2017 [60]	USA	Gender bias	IAT and vignettes	Cross-sectional study with 2 gender IATs on stereotypes (strength and risk taking) and vignettes diffing by gender	503	Physicians	Moderate negative UB on gender stereotype in risk-taking and strong UB on strength IAT against women. No difference for diagnosis but differences in recommendations for testing in vignettes by gender	++/+++/n.a.
Drwecki et al., 2011 [38]	USA	Racial bias	Vignettes	Randomized experimental intervention study with vignettes varying by race (black/white)	40	Nurses	Differences in pain treatment recommendations by race	n.a.
Dy et al., 2015 [61]	USA	Racial bias, gender bias	Vignettes	Cross-sectional computerized vignette study varied by race (black/white) and gender	113	Physicians	No difference in recommendations for surgery in vignettes for race and gender	0
Enea-Drapeau et al., 2012 [62]	France	Disability bias	IAT	Cross-sectional study with 2 disability IAT (Trisomy 21, typical or weakly typical)	55	Health care providers	Implicit negative bias against people with Trisomy 21	n.a.
Fiscella et al., 2021 [63]	USA	Racial bias	IAT and vignettes	Randomized field experiment with standardised patient (black/white) and race IAT (black/white)	90	Physicians	Negative UB on race against black people in IAT. Physicians with stronger UB prescribed less frequently opioids to black patients and those with lower UB less frequently to white patients	n.a.
Galli et al., 2015 [13]	Italy	Disability bias	IAT	Cross-sectional study with disability IAT (wheelchair users/no wheelchair users)	45	Physio-therapists, other health care providers	No UB against wheelchair users	0
Gould et al., 2019 [64]	USA	Disability bias	IAT	Cross-sectional study with disability IAT	290	Genetic counsellors	Strong negative UB against people with disability	+++
Graetz et al., 2021 [65]	USA	Racial bias, SES bias	IAT and vignettes	Prospective study with IAT on race (black/white) and SES (high/low) and case vignettes	105	Health care providers	Strong negative UB against low-SES patients and a moderate negative bias on race (black people). No bias in vignettes	+++/++/0
Guedj et al., 2021 [66]	USA	Racial bias, weight bias	IAT	Cross-sectional online-study with weight IAT and 2 race IATs (black/white and Hispanic/white)	101	Physicians	Strong negative UB against black, Hispanic people and overweight people	+++/+++/+++
Guillermo & Barre-Hemingway 2020 [67]	USA	Racial bias	Vignettes	Cross-sectional study with randomized vignettes varying by race (black/white)	116	Health care providers	No race-based differences in pain estimates nor treatment recommendations	n.a.
Hagiwara et al., 2013, 2016, 2017 [68,69,70]	USA	Racial bias	IAT	Re-analysis of cross-sectional study with race IAT and of racially discordant medical interactions	14	Physicians	Weak negative UB against black people. UB has influence in communi cation style of physicians and inter action when patients reported prior discrimination	+
Haider et al., 2014 [71]	USA	Racial bias, SES bias	IAT and vignettes	Cross-sectional online study with race and SES IATs and vignettes varying by race (black/white) and SES (low/high)	251	Physicians	Moderate negative UB against black and strong negative UB against low SES people in IATs. No differences in treatment decisions in vignettes	++/+++/0
Haider et al., 2015 [72]	USA	Racial bias, SES bias	IAT and vignettes	Cross-sectional online study with race and SES IATs and vignettes varying by race (black/white) and SES (low/high)	215	Physicians	Moderate negative UB against black people, strong UB against people with low SES in IATs. Differences on treatment decisions by race in 3 out of 27 decisions in vignettes	++/+++/n.a.
Haider et al., 2015 [73]	USA	Racial bias, SES bias	IAT and vignettes	Cross-sectional online study with race and SES IATs and vignettes varying by race (back/white) and SES (high/low)	245	Nurses	Moderate negative UB against black people and a strong negative UB against people with low SES in IATs. Differences in treatment decisions in vignettes according to race and SES	++/+++/n.a.
Halvorson et al., 2019 [74]	USA	Weight bias	IAT	Cross-sectional mixed-methods study with weight IAT and semi-structured key informant interviews	28	Physicians, nurses	Moderate to strong negative UB against overweight people	++-+++
Hausmann et al., 2015 [75]	USA	Racial bias	IAT	Cross-sectional online pilot study with race IAT (black/white)	14	Physicians	Moderate negative UB against black people	++
Hirsh et al., 2015 [76]	USA	Racial bias	IAT and vignettes	Cross-sectional study with race IAT and virtual human vignettes varying by race (black/white) and ambiguity of pain	129	Physicians	Moderate negative UB against black people in IAT. No difference in treat-ment options in vignettes with low ambiguity, with high ambiguity decisions varied for white patients not for black	++/n.a.
Hirsh et al., 2020 [77]	USA	Racial bias	IAT and vignettes	Cross-sectional online-study with race IAT and virtual vignettes varying by race (black/white) and addiction history	135	Physicians	Moderate negative UB against black people in IAT. Differences in perceptions about patients’ risks for misuse/abuse by race and past opioid misuse in vignettes	++/n.a.
Hirsh et al., 2014 [78]	USA	Gender bias	Vignettes	Cross-sectional study with computer simulated patient vignettes varying by gender	98	Health care providers	Differences in treatment recommendations by gender	n.a.
Hull et al., 2021 [79]	USA	Racial bias	Vignettes	Cross-sectional online-study with vignette varying by race (black/white)	174	Health care providers	Negative UB against black people in consultation and prescribing behaviour depending on perceived ability of adherence	n.a.
Johnson et al., 2016 [80] and 2017 [81]	USA	Racial bias	IAT	Cross-sectional study with repeated measures (pre- and post- shift) with 2 race IATs (black/ white) (adult and child versions)	91	Physicians	Moderate negative UB against black people both for adults and children pre- and post-shift	++
Kopera et al., 2015 [32]	Poland	Mental illness bias	GNAT	Cross-sectional study with GNAT on mental illness	29	Psychiatrists, psycho-therapists	Negative UB toward people with mental illness	n.a.
Lepièce et al., 2014 [82]	Belgium	Racial bias	Vignettes	Cross-sectional vignette study varying by race (migrant status/ local)	171	Physicians	No differences in medical decisions by ethnicity except prescription of drugs	0/n.a.
Liang et al., 2019 [83]	USA	Diagnosis bias	IAT	Cross-sectional study with 2 IATs on prejudice and stereotypes on cervical cancer vs. ovarian cancer patients	176	Physicians, nurses	Weak negative unconscious prejudice and stereotyping toward cervical cancer patients. Physicians had no UB while nurses did	+/+/0/0
Londono Tobon et al., 2021 [84]	USA	Racial bias	IAT	Cross-sectional online study with 3 race IATs (black/white) related to diagnosis, compliance, and medication	171	Psychiatrists	Weak to moderate negative UB against black people on IATs on dia gnosis, compliance, and medication	+-++/+-++/+-++
Lowe et al., 2020 [85]	USA	Racial bias	(IAT) vignettes	Re-analysis of cross-sectional study with 2 race IATs (black/white and Hispanic/white) of videotaped counselling sessions with simulated patients	60	Genetic counsellors	Results of IATs see Schaa et al., 2015 Slight difference in communication strategies according to race in vignettes no association between communication style and IATs	n.a.
Moskowitz et al., 2012 [35]	USA	Racial bias	Reaction time task with priming	Cross-sectional study with computerized reaction time task with subliminal exposure to black/white photographs	11	Physicians	Unconscious association of certain diseases to black people compared to white people	n.a.
Nash et al., 2014 [86]	UK	Ageism	IAT	Cross-sectional study with ageism IAT	49	Nurses	Strong negative UB towards elderly people	+++
Nymo et al., 2018 [87]	Norway	SES bias	Vignettes	Cross-sectional online-study with vignettes varying by SES (low/neutral).	107	Physicians	Difference in priority of referrals by SES in one of 3 vignettes giving low-SES patients low priority	n.a.
Oliver et al., 2014 [88]	USA	Racial bias	IAT and vignettes	Cross-sectional online study with randomized vignettes varying by race (black/white). Race and cooperativeness IATs	543	Physicians	Moderate negative UB against black people, weak negative UB on their cooperativeness in IATs. No difference in treatment decisions in vignettes according to race	++/+/0
Omori et al., 2012 [42]	Japan	Mental illness bias	IAT	Repeated-measure sstudy before and after contact with mentally ill patients with 2 IATs with different expressions for mental illness and association to “criminal”	51	Physicians	Negative UB against people with mental illness (schizophrenia)	n.a.
Patel et al., 2019 [89]	Italy	Racial bias	IAT and vignettes	Cross-sectional study with brief race IAT and randomized vignettes varying by race (black/white)	57	Dentists	Strong negative UB against black people in IAT. Differences in recommendations for treatment options by race in vignettes	+++/n.a.
Penner et al., 2016 [90]	USA	Racial bias	IAT	Mixed-methods study with race IAT, recorded physician–patient interactions and follow-up interviews	18	Physicians	Weak negative UB against black people	+
Puumala et al., 2016 [91]	USA	Racial bias	IAT and vignettes	Cross-sectional online study with vignettes. Two race IATs (American Native/white), child and adult versions	101	Physicians, nurses	Moderate negative UB against American Native people on IAT on child and adult versions. Only little differences in treatment recommen-dations in vignettes	++/++/n.a.
Robstad et al., 2018 [92]	Norway	Weight bias	IAT and vignettes	Cross-sectional online pilot study with 2 weight IATs on attitudes and stereotypes and vignettes varying by weight	30	Nurses	Strong negative UB on attitude and moderate on stereotypes against overweight people, no difference in behavioural intention in vignettes	+++/++/0
Robstad et al., 2019 [93]	Norway	Weight bias	IAT and vignettes	Cross-sectional study with two weight IATs on attitudes and stereotypes and vignettes varying by weight	159	Nurses	Strong negative UB against overweight people in both IAT, no difference in behavioural intention in vignettes	+++/+++/0
Rojas et al., 2017 [94]	USA	Racial bias	Vignettes	Cross-sectional online study with vignettes varying by race (black/white)	342	Physicians	No statistically significant differences in suspicion for abuse-related injury based on race of child	0
Sabin & Greenwald 2012 [95]	USA	Racial bias	IAT and vignettes	Cross-sectional online study with 3 race IATs on attitude, compliance, and stereotypes and vignettes of child patients varying by race (black/white)	86	Physicians	Weak negative UB on attitude, moderate on compliance and stereo type against black people on IATs. Differences in treatment recommen dations by race in 1 out of 4 vignettes	+/++/++/n.a.
Sabin et al., 2012 [96]	USA/Global	Weight bias	IAT	Re-analysis of data from weight IAT from Project Implicit (2006–2010)	2284	Physicians	Strong negative UB against overweight people	+++
Sabin et al., 2015 [97]	USA	Weight bias, racial bias	IAT	Cross-sectional online pilot study with weight and race IATs on (Native American/white)	75	Physicians, nurses, physician assistants	Strong negative UB against overweight people, weak negative UB against Native Americans	+++/+
Sabin et al., 2015 [98]	USA	LGBTQ bias	IAT	Re-analysis of data from LGB IAT from Project Implicit (2006–2012)	18,983	Physicians, nurses, mental health providers, other health care providers	Weak to moderate negative UB against homosexual people	+-++
Sandhu et al., 2019 [99]	Canada	Mental illness bias	IAT	Cross-sectional online study with IAT on mental illness	538	Psychiatrists	No UB against people with mental illness	0
Schaa et al., 2015 [100]	USA	Racial bias	IAT	Mixed-methods study with cross-sectional online survey with IAT on race (black/white) and with re-analysis of data on patient–physician interaction	60	Genetic counsellors	Moderate negative UB against black people. Differences in communi cation style during counselling sessions according to race	++/n.a.
Schoenberg et al., 2019 [101]	USA	Racial bias, gender bias	Vignettes	Cross-sectional online pilot study with vignettes varying by race (skin tone) and gender	80	Physicians	Treatment options differed by skin colour and gender according to stereotypes	n.a.
Schroyen et al., 2016 [102]	Belgium	Ageism	Vignettes	Cross-sectional study with ran domized vignettes varying by age	76	Nurses	Negative UB by age that increases as age of patient increases	n.a.
Setchell et al., 2014 [103]	Australia	Weight bias	Vignettes	Cross-sectional online survey with vignettes differing in body mass index	265	Physio-therapists	Minimal statistically not significant differences in treatment options by weight of patients	0
Shapiro et al., 2018 [36]	USA	Racial bias, SES bias	IAT and vignettes	Cross-sectional study with race and SES IATs and vignettes varying by race (black/white) and SES.	971/549/530	Physicians	Moderate negative UB against black women, strong negative UB against women with low SES in IATs. No differences of treatment recommen dations by race but by SES in vignettes	++/+++/0/ n.a.
Siegelman et al., 2016 [104]	USA	Racial bias	IAT and vignettes	Cross-sectional study with race IAT and vignettes varying by race (black/white)	57	Physicians	No results for IAT published and some difference in pain treatment by race in vignettes	n.a
Stepanikova 2012 [105]	USA	Racial bias	Vignettes	Cross-sectional online vignette study with/without racial priming varying by race (black/Hispanic/ white) and cognitive load.	81	Physicians	UB in diagnosis and referral against black people and less so towards Hispanic people under time pressure, less so if there is no time pressure	n.a.
Sukhera et al., 2018 [106] and 2019 [17]	Canada	Mental illness bias	IAT	Cross-sectional mixed-methods study with IAT on dangerous ness of mental illness and semi-structured interviews about results	31	Psychiatrists, Nurses	32% had a negative UB on dangerousness of mentally ill people and 55% on physically ill people, and 13% had no UB	n.a.
Tajeu et al., 2018 [107]	USA	Racial bias	IAT	Cross-sectional online-study with race IAT (black/white)	107	Physicians, nurses, other health care providers	Moderate negative UB against black people	++
Tucker Edmonds et al., 2017 [108]	USA	Racial bias	Vignettes	Cross-sectional pilot study with self-administered vignettes varying by race (black/white)	77	Physicians, nurses	Differences in treatment options by race	n.a.
Vaimberg et al., 2021 [40]	USA	Disability bias	IAT and vignettes	Repeated measure online pilot study (pre- and post-intervention) with disability IAT (pre-intervention) and vignettes (with/without physical disability) pre- and post-intervention	335	Physicians, nurses, genetic counsellors, other health care providers	Negative UB on disabled people (84% of respon-dents), UB influencing clinical recommen dations. After intervention UB decreased, recommendations changed	n.a.
VanPuymbrouck et al., 2020 [109]	Global	Disability bias	IAT	Re-analysis of data from disability IAT from Project Implicit (2004–2017)	25,006	Physicians, dentists, nurses, occupational- and physio-therapists, other health care providers	Moderate negative UB against disabled people	++
Walden et al., 2020 [110]	USA	Disability bias	IAT	Cross-sectional study with disability IAT (stuttering/normal speech)	15	Speech–language pathologists	Moderate negative UB against people that stutter	++
Wandner et al., 2014 [111]	USA	Racial bias, gender bias, ageism	Vignettes	Cross-sectional online vignette study with human virtual profiles varying by race, gender, and age	193	Physicians, nurses	Differences in pain assessment by race and gender, not by age. Male and black people were rated to have more pain	n.a./0
Welch et al., 2012 [112]	USA	Gender bias	Vignettes	Cross-sectional mixed-methods study with video vignette varying by gender with/without cognitive priming, think-aloud interviews	256	Physicians	Differences in treatment options by gender	n.a.
Welch et al., 2015 [113]	USA	Mental illness bias	Vignettes	Cross-sectional video-based vignette study varying by symptoms of mental and physical illness	256	Physicians	Negative stereotypes against people with mental illness	n.a.
Wijayatunga et al., 2021 [43]	USA	Weight bias	IAT	Randomized online study with repeated measures (pre-/post-intervention, 1-month follow-up) with weight IAT taken 3 times	147	Dieticians	Negative UB against overweight people at all 3 measurement points	n.a.
Wittlin et al., 2019 [114]	USA	LGBTQ bias	IAT	Longitudinal study with 3 repeated measures with IAT on lesbian and gay people	1155	Physicians	Weak negative UB against lesbian and gay people at both time points	+/1
Zestcott et al., 2021 [115]	USA	Racial bias	IAT	Cross-sectional online-study with 2 IAT on race (American Native/ white) on prejudice and stereotypes	111	Physicians	Moderate negative UB toward American Natives in attitudes and stereotypes	++/++

Legend: + weak, +-++: weak to moderate, ++ moderate, ++-+++ moderate to strong, +++ strong unconscious bias in IAT; n.a., strength of bias not applicable; 0, no unconscious bias found.

### 3.4. Biases

Of the 81 studies included, 67 assessed one bias, 12 studies two biases, and 2 studies three biases. In total, 97 biases were investigated (see Table 2). Racial bias was assessed against African Americans or “black” people in 41 studies, against Hispanic or Latino Americans in 5 studies, and against Native American people in three studies relative to “white” people. Four studies assessed more than one race. One study [82] assessed the biases of local (European) general practitioners against individuals with migrant status compared to local people using a foreign (Moroccan) vs. a typical local (Belgian) name. No study investigated bias against Asian people or people from other racial or ethnic backgrounds. The studies on racial bias were conducted predominantly in the United States. Only two studies that addressed racial bias were conducted in Europe: one on race (black/white) [89] and one on migrants (migrant/local) [82]. 

Racial bias was assessed with 14 studies with implicit measures, in 16 studies with vignettes and in 16 studies with an implicit measure and vignettes. Amongst the 30 studies using an implicit measure, 28 used an IAT, and 1 each used a reaction time task [35] and the AMP [33]. All showed an unconscious racial bias against the minority group assessed, even though participants from minority groups might have had a lower unconscious bias, no unconscious bias, or an unconscious bias against white people (e.g., [36]). The 32 studies using a vignette showed in 22 cases a difference in treatment decisions or recommendations, while in ten cases there were no differences. In five studies, implicit measures found an unconscious bias while vignettes showed none. 

While in almost all studies bias was against a minority, two studies found a bias against a majority: One study using the vignette method [108] found that white patients were more suspected to divert pain medication compared to black patients. Another vignette study [76] found that white patients received less consistent care when the vignette was manipulated in such a way that clinical ambiguity (pain etiology, congruence of facial expression, and pain report) was high. 

The racial background of study participants reflected the composition of health professionals in the respective countries: In the United States, they were predominantly Caucasian/white with minorities of African American/black, Asian/Pacific Islander, Hispanic/Latino, and Native American people, as well as people identifying as mixed race. In the European studies, participants were Europeans. 

The second-most studied bias was gender bias, with 11 studies, all of which used vignettes to assess gender bias. In addition, one study also used an IAT [60]. Nine studies showed an unconscious bias against women, while two studies found no differences according to gender in vignettes. The IAT showed a strong unconscious bias against women and a moderate negative unconscious bias for the stereotype of women and risk-taking.

Weight bias was examined in ten studies; seven studies used an IAT, one study used vignettes only, and two studies used both methods. Of the ten studies, eight showed an unconscious bias while two studies found no [53] or only a minimal [103] difference in treatment decisions. Six studies using an IAT showed a moderate to strong unconscious bias while two studies did not report on the strength of unconscious bias [43,47]. The vignette study showed only a minimal difference in treatment options [103], and of the two studies using an IAT and vignettes, one found a difference in treatment recommendation but not in treatment options between normal-weight patients and overweight patients. The other study found no difference between the two groups in treatment intentions [93], while both IAT studies showed a strong unconscious bias. 

Socio-economic status (SES) bias was addressed in nine studies. Of these, seven studies used both an IAT and vignettes, while two studies were based on vignettes only. Of the seven studies using both an IAT and vignettes, all showed an unconscious bias in the IAT ranging from moderate to strong. In the vignettes, however, two studies did not find a difference in treatment decisions, and five did. Of the two studies which used vignettes, only one showed a difference in treatment decisions [87] and one did not find a difference in time spent on diagnosis or in accuracy [45].

Mental illness bias was addressed in seven studies using the general term “mental illness” three times, schizophrenia twice, depression one time, and the combination of mentally ill and dangerous once. Six studies assessed the unconscious bias against mentally ill people with an implicit measure: the IAT was used six times and the GNAT was used once. One study used vignettes. While five studies using an implicit measure found an unconscious bias, one did not [93]. The vignette study found an unconscious bias. 

In seven studies, disability bias was examined. In three studies, the term “disability” was used, and one study each assessed the bias against people with Trisomy 21, people in wheelchairs, people with cleft lips, and people who stutter. In six studies an IAT was used. Of these, only one study found no unconscious bias (against wheelchair users) [13]. The study which used both an IAT and vignettes showed an unconscious bias in the IAT and a difference in recommendations according to disability [40]. 

Ageism was examined in four studies. The IAT was used once [86], showing a strong unconscious bias against elderly people. In three studies, vignettes were used, all of which found an implicit bias against the elderly, two of which showed differences in pain ratings [49,111], and one showed decreasing support for oncology treatment as the age of patients increased [102].

Little research was conducted on LGBTQ bias, which was assessed in two studies, both using the IAT and both showing a weak to moderate implicit bias against lesbian and gay people [98,114]. 

One study [83] tested whether there was a bias against certain diagnoses. For this study, they compared cervical cancer with ovarian cancer in two IATs. Both prejudice and stereotypes were found against patients with cervical cancer.

While most studies assessed unconscious bias towards adults, four studies tested unconscious bias against children. Two studies compared the IAT child version to the adult version. In one of these studies [91], two versions of the race IAT were compared using Caucasian and Hispanic pictures of adults for one test and pictures of children for the other. The other study [81] did the same using pictures of Caucasian and African American people. In both studies, there was an unconscious bias against Hispanic and African American people, but there was no significant difference for the child vs. adult versions of the IAT. The residual two studies used vignettes with children or youth. One study [95] assessed treatment recommendation using child vignettes and the other [67] used vignettes with adolescents to estimate pain ratings. In both studies, differences in treatment decisions between races were found. However, no comparisons to adults were made. 

### 3.5. Geographical Distribution

Of the 81 studies included, 79 were from 15 different countries and two studies were global in scope and used large datasets from Project Implicit [116]. Most studies were conducted in North America (*n* = 60), followed by Europe (*n* = 13), Australia (*n* = 3), Asia (*n* = 2) and Africa (*n* = 1) (see Table 3).

### 3.6. Involved Health Professions 

As Table 4 shows, the majority of studies involved physicians (*n* = 51), followed by nurses (*n* = 20). Other health professions were studied to a much lesser extent. They were not of focal interest but were specified as components under “health care providers” or “other health care providers”. 

In Table 5, we combine the information on health profession with that on bias types. It shows that bias on race, weight, and disability were studied amongst a wider range of health professionals than other biases, such as ageism or gender bias.

## 4. Discussion

The current review retrieved 87 records from 81 studies on unconscious bias. The findings indicate that there are 87 studies from around the globe that addressed different types of biases and used different methods and samples. The results suggest that unconscious biases seem to be widespread amongst health professionals. Whereas almost all implicit measure studies observed implicit biases in health professionals (53 out of 55), 75% of vignette studies reported implicit biases (33 out of 44). The implicit biases thereby include race/ethnicity, gender, SES, age, mental illness, LGBTQ people, weight, and disability. 

Most studies in this review examined and found racial biases. Even though racial bias was assessed in 46 studies, most of these studies examined racial bias against African American/black people compared to Caucasian/white people. Other races or ethnic groups have rarely been assessed. Therefore, it is unclear how widespread or how strong racial bias against other ethnic groups is and how this racial bias and stereotypes manifest themselves in other regions of the world. Apart from racial bias, other biases like gender bias, ageism, LGBTQ bias, disability bias, and diagnosis bias have received little attention, even though these biases concern a considerable part of the population. 

For most biases, there have been too few studies to claim strong evidence. Moreover, it would be relevant to compare the size, patterns, and interactions of biases against groups and outcomes to better understand how they work and how they can be overcome. Some stereotypes might be more salient and more relevant than others or some stereotypes might be an indicator of further stereotypes. For instance, all studies in our review that assessed both SES bias and racial bias showed that the bias against low SES is greater than that against belonging to an ethnic minority [36,46,65,71,72,73]. In future research, the phenomenon of intersectionality should be given more attention. Intersectionality is when an individual belongs to more than one minority group and might be subject to several unconscious biases, e.g., being an African American woman. Even though 14 studies included in this review examined more than one bias, and in several studies, intersectionality might have been observed, the results are inconclusive as they either did not find or report on intersectionality. Therefore, further research on intersectionality is needed. 

Considering where the research has been conducted, almost 75% of the studies in our review were conducted in the United States (*n* = 58). In contrast, the topic has been as widely examined by researchers in Europe or elsewhere in the world. Evidence in this review shows that there is also unconscious bias in other parts of the world. However, studies are too sparse and too heterogeneous concerning other biases to claim strong evidence. This is especially true because there are major structural and cultural differences between the regions, and the results are therefore not easily transferable to other countries or regions. More research is needed in other world regions to find out which unconscious biases prevail to what extent and how they can be addressed under the given circumstances to prevent adverse outcomes. 

Although implicit biases were found in all health professions, most studies focused on physicians (*n* = 51) or nurses (*n* = 20), while other health professionals were only sporadically studied or subsumed under “health care providers” or “other health care providers”. Two studies in our review found differences between professions. Crapanzano and colleagues [59] observed that physicians in general had an unconscious bias towards mentally ill people, whereas psychiatrists did not. Moreover, Liang and colleagues [83] found that nurses had an unconscious bias towards patients with cervical cancer, while physicians did not. These findings suggest that further research on potential differences between health professions would be promising, as it might provide insights into the origins of implicit biases and their perpetuation. For example, it could help to identify differences in the socialization of the profession and perhaps the passing on of stereotypes and biases, or differences in professional training and practice that are relevant to biases. It would also be interesting to compare care models and differences in patient-centeredness, as well as institutional factors, in relation to unconscious bias to find out whether care practices and institutional settings can mitigate unconscious bias.

In this respect, the three intervention studies [33,38,40] that used empathy-including methods produced interesting results. They showed a reduction in implicit bias when health professionals were encouraged to consider the perspectives of their patients. Additionally, the study [41] that used real-time information exchange resulted in unbiased treatment recommendations. Further intervention studies should explore how unconscious bias can be reduced. Furthermore, evidence-based interventions should be part of any educational program for health professionals. 

In conclusion, the present review confirms the persistence of an unconscious bias amongst health professionals that was observed in earlier research [10]. Health professionals seem to apply stereotypes to patients’ superficial characteristics. They then derive their expectations towards patients from those characteristics. These expectations in turn influence their choice of recommendations for treatments and prescriptions. In the study of Hull et al. [79], the expectation of the physicians that a black patient would adhere less to an HIV pre-exposure prophylaxis (PrEP) medication compared to a white patient led them to be less willing to discuss the prophylaxis in a consultation. In the study of Lund [53], no difference in treatment decisions was observed in a vignette study that manipulated body mass index and sex. However, general practitioners expected less compliance from obese male patients than obese female patients or normal-weight patients and therefore would not recommend a headache diary to male patients. In line with this assumed process, by which implicit biases impact expectations and behavior, are the results of Cooper et al. [39] who investigated compliance stereotyping in a race and compliance IAT. They found that while black people were strongly associated with words like reluctant, apathetic, and lax, white people were more strongly associated with words like willing and reliable, thus depicting the latter as more adherent. The same study also found a link between implicit bias and communication style. The greater the implicit bias, the greater the verbal dominance in recorded visits of black patients, and the more negative ratings from patients compared to white patients. These findings can also be seen as evidence of how unconscious stereotypes form expectations, and of how they translate into communication and possible consequences for patients. 

We found that multiple disciplines have contributed to this area of research. Often, these disciplines do not recognize each other, as has been pointed out in an earlier review by Fitzgerald and Hurst [10]. Furthermore, different terminologies are used to discuss the same or very similar concepts, such as “attitudes”, “stigmatizing attitudes”, “prejudice”, “stereotypes”, or “bias”, among others. When reading the abstract, it was often unclear what phenomenon was being studied and how it was operationalized. It was therefore difficult to conclude whether it was an unconscious or explicit bias that was being studied. In addition, many articles lacked a definition of terminology or concepts used, which makes it difficult to compile previous research results on unconscious bias. More uniform definitions would facilitate the synopsis of evidence and the progression of the research field. 

We found that there were many different approaches to exploring conscious and unconscious bias. For this review, many studies had to be excluded as their used measures used did not consider unconscious bias. Rather, explicit bias and structural discrimination were measured, or explicit and implicit bias could not be distinguished. In this review, we included only empirical studies that used instruments that to our knowledge can measure unconscious bias. This was either vignette studies or implicit measures. 

Regarding implicit measures, 55 out of 81 studies included in this review used an implicit measure. The most often (*n* = 52) used approach was the “Implicit Association Test” (IAT) [18]. Other implicit measures, each used once, were the Go/No-Go Association Task (GNAT) [19], the Misattribution Procedure [34], and a reaction time task. These have the advantage of internal validity but have been criticized for their lack of external validity [117]. The IAT has shown validity in predicting bias (prejudice and stereotyping) more accurately than self-reports [118]. However, there is a debate regarding whether these implicit measures can predict behavior [119,120,121] and that in situations in which deliberate decisions are made, explicit measures of attitudes and behavioral intentions are more reliable than implicit measures, which in turn are better at predicting spontaneous reactions [122]. This might explain the discrepancy between the findings of unconscious bias using implicit measures and vignettes. In the present review, studies that used an IAT and vignettes arrived in 1/3 of cases at contradictory results. Both deliberate as well as spontaneous decisions are common in the daily life of health professionals. 

Regarding vignette studies, they manipulate the variable of relevance, such as race or gender, and measure responses to fictional patients, while the rest of the portrayed clinical case is identical. This is a rather indirect approach to measure unconscious bias and may seem imprecise. However, the method has external validity, as vignettes are frequently used in the training of health professionals and in patient case descriptions, and may be similar to case reports in medical practice [27]. The results of these vignette studies included in this review are somewhat complex and heterogeneous because the study design often provided several different vignettes or vignettes with multiple options to choose from. In total, 33 out of 44 studies showed that a change in one variable in a vignette, such as gender or weight, leads to at least some differences in judgements, treatment recommendations, and decisions, some of which were considered as unsafe for the patient (e.g., [41]). Other differences were minor, such as the communication methods used (e.g., [85]). In many cases, there was no difference at all. These differences may reflect real unconscious biases but may also reflect learned practices built on stereotypes. However, tests with vignettes are a good opportunity to compare, reflect on, and question attitudes, stereotypes, intentions, and treatment routines. 

Future research should address the differences between the different measures and their distinct predictability in greater detail and find out which situations in the everyday life of health professionals are particularly prone to unconscious bias. 

This review has several limitations. It only covers the period of 2011 to 2021, and studies published earlier were not included in this review. Thereby, we may have excluded studies that would have provided more conclusive insight into the unconscious bias in other health professions and regions, or which might have given more insight on those biases studied only seldomly (e.g., ageism). However, as the present review builds on the one by Fitzgerald et al. [10], prior research is indirectly considered in our discussion. 

Because we included only studies that, to our knowledge, measured unconscious bias, we had to exclude a great number of studies. These included studies with a qualitative approach and studies in which explicit and unconscious bias could not be separated. These studies might have provided more insight into how unconscious bias is perceived by patients or how health professionals try to mitigate their unconscious bias. The focus of our review, however, was different. A qualitative review could provide further insights in this regard to better understand the mechanisms and consequences of unconscious biases. However, we encounter the difficulty that implicit and explicit bias are two distinct phenomena that must be considered separately from each other. 

Publication bias most likely occurred amongst studies evaluating unconscious bias in health professionals. Studies that found unconscious bias in health professionals might have been more likely to be published than if their results were inconclusive or indifferent. Therefore, it is difficult to estimate the prevalence of unconscious bias among health professionals. The combined results show that unconscious bias is widespread among health professionals. 

As we did not systematically assess the quality of the studies in our review, the quality of the included studies should be interpreted with caution. In some cases, studies did not provide results for the IAT with the d-scores suggested by Greenwald and colleagues [123]. It is, therefore, difficult to compare or compile the results. More rigor in the reporting of results would be beneficial for the research field. 

This review shows that, apart from in the United States, there has been only little research on unconscious bias. Given that most studies have been conducted in the United States, the transferability to Europe or other regions of the world is limited due to cultural differences concerning minorities and the awareness of health disparities and unconscious bias, in addition to differences in the health care systems and society. This is particularly the case with racial bias, which has been investigated almost exclusively in the United States in the context of black people in comparison to white people. Research on other minorities or bias towards immigrants, asylum seekers, and refugees is very sparse. It would be important to learn which minority groups are specifically concerned in which country, how different biases are generally associated with each other, and how they are linked within health care professionals. 

A further research gap lays in the type of biases assessed: research on gender bias, SES bias, disability bias, ageism, and weight bias is sparse, even though these biases concern most societies around the world. 

The unconscious bias of physicians and nurses has been studied most frequently. Physical and occupational therapists, midwives, dieticians, physician associates, pharmacists, and other health care professionals are other decision makers in terms of patient care. Therefore, these professionals should also be included in further studies of unconscious bias. 

In this review, only five studies tested an intervention. Further studies are needed to better understand how unconscious bias can be mitigated and to ensure good health care for all people.

## 5. Conclusions

The majority of health professionals assessed in the studies included in this review have an unconscious bias against certain groups of patients. There are some challenges in the conceptualization of unconscious bias, the terminology used, and the use of instruments. Despite these challenges, methods to examine unconscious bias are available and can be used. The relationship between unconscious bias and health disparities remains somewhat unclear. There are considerable research gaps, with little research conducted outside the United States and on other biases besides racial bias. Additionally, health care professionals other than physicians and nurses have been seldom studied. Given the pervasiveness of unconscious biases, and the potential discrimination that might arise as a consequence, more research on the consequences and generalizability of biases, the emergence processes, and interventions to reduce unconscious biases is strongly called for.

## Figures and Tables

**Figure 1 ijerph-20-06569-f001:**
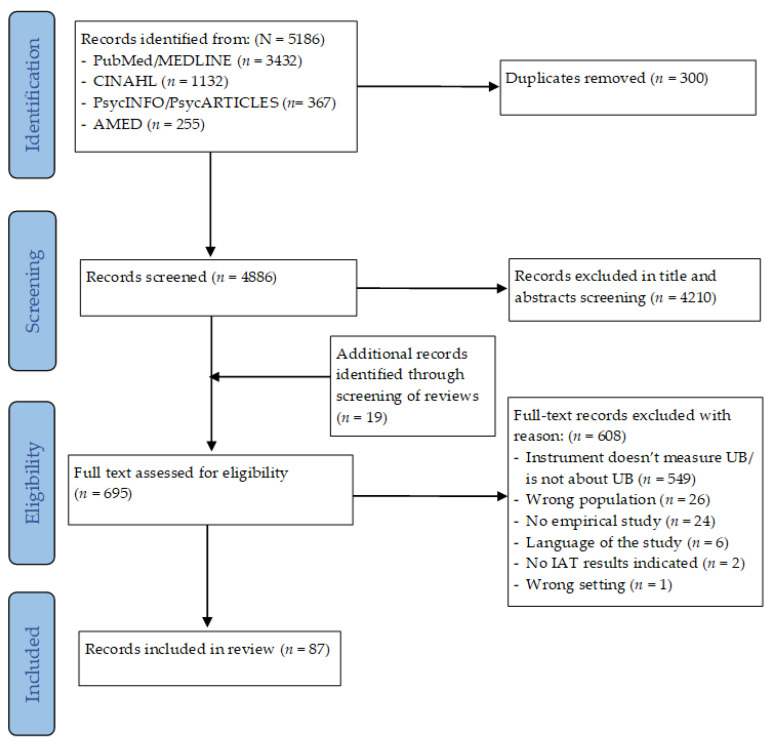
PRISMA flow diagram of scoping review.

**Table 2 ijerph-20-06569-t002:** Number of assessed bias by type (focus).

Bias	Number of Studies
Racial bias	46
Gender bias	11
Weight bias	10
SES bias	9
Mental illness bias	7
Disability bias	7
Ageism	4
LGBTQ bias	2
Diagnosis bias (cervical cancer)	1
Total	97

Note that in some studies, multiple biases were assessed in the same study.

**Table 3 ijerph-20-06569-t003:** Type of bias, number of biases assessed per bias, and total number of biases assessed per country.

Country	Type of Bias	Number of Biases Assessed per Bias	Total Number of Biases Assessed per Country
Australia (*n* = 3)	Weight bias Mental illness bias	2 1	3
Belgium (*n* = 2)	Ageism Racial bias	1 1	2
Canada (*n* = 2)	Mental illness bias	2	2
Denmark (*n* = 1)	Weight bias	1	1
France (*n* = 1)	Disability bias	1	1
Italy (*n* = 2)	Disability bias Racial bias	1 1	2
Japan (*n* = 1)	Mental illness bias	1	1
Kenya (*n* = 1)	SES bias	1	1
Norway (*n* = 3)	Weight bias SES bias	2 1	3
Poland (*n* = 1)	Mental illness bias	1	1
Portugal (*n* = 1)	Gender bias	1	1
Saudia Arabia (*n* = 1)	SES bias	1	1
Sweden (*n* = 1)	Gender bias	1	1
UK (*n* = 1)	Ageism	1	1
USA (*n* = 58)	Racial bias	44	
	Gender bias	9	
	SES bias	6	
	Weight bias	4	
	Disability bias	4	
	LGBTQ bias	2	
	Mental illness bias	2	
	Ageism	2	
	Diagnosis bias	1	74
Global (*n* = 2)	Disability bias	1	
	Weight bias	1	2

**Table 4 ijerph-20-06569-t004:** Number of biases by profession.

Bias	Number of Studies
Physicians	51
Nurses	20
Health care providers (not specified)	12
Psychiatrists	5
Dentists	4
Genetic counsellors	4
Physiotherapists	3
Mental health providers (not specified)	2
Physician assistants	1
Speech–language pathologists	1
Dieticians	1
Diagnostic radiographers	1
Occupational therapists	1
Midwives	1
Psychotherapists	1
Other support staff	1

Note that in some studies, multiple professions were assessed.

**Table 5 ijerph-20-06569-t005:** Number and type of biases assessed by health profession.

	Physicians	Nurses	Health Care Providers (Not Specified)	Dentists	Psychiatrists	Genetic Counselors	Physiotherapists	Rehabilitation Staff (Not Specified)	Physician Assistants	Mental Health Providers (Not Specified)	Dietitians	Occupational Therapists	Speech–Language Pathologists	Diagnostic Radiographers	Midwives	Psychotherapists	Other Support Staff
Race	34	8	6	2	1	2		1	1								
Gender	8	3	1	1				1									
SES	7	2	1												1		1
Weight	5	4					1		1		1			1			
Mental illness	3	1			4					1						1	
Disability	2	2	4	2		2	2					1	1				
LGBTQ	2	1	1							1							
Ageism	2	3		1													
Diagnosis	1	1															
Total	64	25	13	6	5	4	3	2	2	2	1	1	1	1	1	1	1

Note that in some studies, multiple professions and multiple biases were assessed.

## Data Availability

No new data were created or analyzed in this study. Data sharing is not applicable to this article.

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
