# Peer review of "Unconscious Bias among Health Professionals: A Scoping Review"

_ijerph, 2023, doi:10.3390/ijerph20166569_

Round 1
Reviewer 1 Report
Review Report: ijerph-2480502-peer-review-v1
Title: Unconscious bias among health professionals. A scoping review
Overall: Overall it is a good article.
Title: The topic is unique and well thought out.
Abstract: It is good. Keywords are mentioned.
Background:
The background is aligning with the objectives.
It has clearly mentioned what did prompt to conduct the study, which is good.
Method & Design:
Although the authors mention the use of PRISMA guidelines and many of its components are not actually described within the manuscript.
PCC (Population (or participants)/Concept/Context) framework used is quite commendable with Appendix A: Table with search strategy
Provide justification for selecting “Implicit Association Test”, the “Go/No-Go Association Task” and vignette studies. and justification of period range selected.
Was there an update search or an email alert set for the databases?
Any search of registries or grey literature?
Results:
Results are presented quite comprehensive. The authors could have reported percentage along with number of studies while presenting the result.
Geographical distribution of studies could have been presented with a 3D World map.
Further is it possible to make an inference about biases among physicians versus other health professionals. That will further help in establishing where it is more prevalent.
Discussion:
The discussion is good and in alignment with the objectives and findings.
Conclusion
The conclusion is acceptable. Further is it possible to make an inference about biases among physicians versus other health professionals. That will further help in establishing where it is more prevalent.
Limitations of study
limitation of the paper has been included.
Informed Consent Statement: A statement from authors is missing.
Data Availability Statement: A statement from authors is missing.
Author Response
Response to Reviewer
Dear Reviewer
Thank you for your thorough reading and the useful comments and suggestions to our article on unconscious bias. Below you find your comments and recommendations and our responses. We also submit a version of the manuscript in track changes mode.
Language:
English language is fine.
Response: Thanks. The text was edited by a highly professional academic writer from the US who is still active in research.
Overall:
Overall, it is a good article.
Response: Thanks for this positive overall assessment of our work.
Title:
The topic is unique and well thought out.
Response: We appreciate this positive feedback.
Abstract:
It is good. Keywords are mentioned.
Response: Thanks for this positive evaluation.
Background:
The background is aligning with the objectives. It has clearly mentioned what did prompt to conduct the study, which is good.
Response: Thank you for the positive evaluation of the alignment and reasoning.
Method & Design:
Although the authors mention the use of PRISMA guidelines, many of its components are not actually described within the manuscript.
Response: Thank you for this comment. We explored the possibility of conducting a systematic review that would make use of the PRISMA guidelines but ultimately decided to conduct a scoping review as it allows including a broader range of studies, to evaluate the scope of available research, and to identify research gaps.
Although a PRISMA-ScR extension has been published as well by Tricco and colleagues in 2018, we followed the more recent JBI International Scientific Committee’s guide on conducting scoping reviews that focuses on new updates to the approach and development of the Preferred Reporting Items for Systematic Reviews and Meta-Analyses extension for Scoping Reviews (the PRISMA-ScR). We double-checked that the central components of this guide are included in the manuscript and added the following specification [in italics] in the methods section (p.3) “we followed the JBI International Scientific Committee’s guide on conducting scoping reviews that builds on the PRISMA-ScR guideline [22]”. Moreover, we renamed the title of the diagram on page 4 to “PRISMA Flow diagram of the scoping review” to clarify our use of the respective PRISMA tool. We hope that that these adaptions resolve the issues concerning the use of the PRISMA resources.
PCC (Population (or participants)/Concept/Context) framework used is quite commendable with Appendix A: Table with search strategy
Response: Thanks for this positive feedback on the alignment of the PCC framework and the Table in Appendix A. In order to further explicate the link between the PCC framework and the contents of the Table, we now added the term “Concept” to the previous description (“Phenomenon of interest: Unconscious Bias/ Implicit Bias”). Now the row descriptions in the left column of the table read: «Population», «Concept», and «Context».
Provide justification for selecting “Implicit Association Test”, the “Go/No-Go Association Task” and vignette studies.
Response: In order to clearly differentiate between conscious/explicit and unconscious/implicit bias one needs to argue which instruments are apt to detect unconscious bias. From the validity of the concepts measured by IAT, the Go/No-Go Association Task and some vignette studies it becomes clear that they measure unconscious biases. Therefore, we focused on these tests and modes of eliciting unconscious biases.
Provide justification of period range selected.
Fitzgerald and Hurst who published their results in 2017 ended their search in March of 2013. It was our intention to cover the period after 2013 and have some overlap to not miss out on studies that were published in 2013 but were not included in the review by Fitzgerald and Hurst.
Was there an update search or an email alert set for the databases?
Response: No, we would have liked to do both. However, given the time limitations of the PhD, we were faced with the decision whether we either wanted to perform a) a very comprehensive search to ensure that we detect all relevant studies in this very heterogeneous research field but only up to the time of the start our search or b) a search with a more limited focus that we can update over the time of the review process. We chose the former and can now provide a review that covers more than 4’200 records whose title and abstract were screened as well as almost 700 screened full texts for the time that we cover. Given that there have been no breakthrough innovations in the research on unconscious bias since then, we think that there is value in the reported results and hope that this procedure and its’ accompanying limitations are acceptable.
Any search of registries or grey literature?
Response: Although we encountered few project reports that mentioned that they conducted an IAT, they were lacking sufficient information on the methodological details as well as on the analyses. Therefore, we did include a search of the grey literature in our review.
Regarding the registries, we expected that relevant publications that are based on registry data would also surface when using our search string. In line with assumption, we found and included a study in our review that used a nationally representative sample of registered dietitians [44]. Together, we would like to argue that not including searches in grey literature and registries does not compromise the quality of the present review as it is important to limit the review to unconscious bias measurements that are documented well enough to ensure the transparency and reproducibility of the respective research.
Results:
Results are presented quite comprehensive. The authors could have reported percentage along with number of studies while presenting the result.
Response: In instances where there are small numbers, we think percentages do not provide much additional value. Moreover, we do not intend to quantify the results in the sense of classical methods when presenting results of systematic searches such as a forest plots.
Geographical distribution of studies could have been presented with a 3D World map.
Response: The studies are mainly from North America and Europe. A 3D World map would not yield much additional information.
Further is it possible to make an inference about biases among physicians versus other health professionals. That will further help in establishing where it is more prevalent.
Response: We did not make a statement on purpose. The body of studies included does not allow an assessment of whether unconscious biases are more widespread among physicians versus other health professionals. It is our plan to initiate such a study using quantitative methods by including a representative sample of physicians and other health professionals in various European countries.
Discussion:
The discussion is good and in alignment with the objectives and findings.
Response: Thanks for this reassuring feedback on the stringency of the manuscript.
Conclusion:
The conclusion is acceptable. Further is it possible to make an inference about biases among physicians versus other health professionals. That will further help in establishing where it is more prevalent.
Response: See our comment above on the situation of studies including physicians and other health professionals.
Limitations of study:
Limitation of the paper has been included.
Response: Thank you for valuing our efforts to be transparent on the strength and limitations of the study.
Informed Consent Statement:
A statement from authors is missing.
Response: As we did not perform a study ourselves, an informed consent statement cannot be gathered. If we misunderstand anything, please let us know. We will exclude the statement according to the journal’s requirements as we did not directly involve humans; they were included in the studies that we retrieved in this scoping review.
Data Availability Statement: A statement from authors is missing.
Response: As we used only published literature from publicly available journals and publicly accessible bibliographic databases such as Pubmed or CINAHL and we included the extraction table in the manuscript, we are not exactly sure what to do. We are familiar with data availability statements in the case of original research with patients or any kind of groups from the general population or additional data that is available in the quality evaluation section of systematic reviews but in our view, this does not apply to scoping reviews. We followed the suggested data availability statements available online (https://www.mdpi.com/ethics) and added the following sentence in the manuscript (p. 24): “No new data were created or analyzed in this study. Data sharing is not applicable to this article.”
Reviewer 2 Report
Lines 80-81: The sentence here does not read well. Please reconstruct it.
Lines 93-96: I commend the authors for mentioning the difference between their study and FitzGerald and Hurst.
Lines 100-101: FitzGerald and Hurst did a systematic review. It would have been better if the authors did systematic review too in order to be able to compare the two studies appropriately. However, the scoping review is fine.
Line 180: This is not clear. A prospective design is also a longitudinal design, so why n=1 and n=1 respectively?
Lines 209-210: What could have led to this difference in outcome?
Lines 239-240: The authors should re-write this sentence because it is not clear.
Lines 321-324:This means that both the vignettes and IAT are not in the same direction. The question is, which of them is reliable?
Lines 390-391:I am not sure that the fact that few studies were conducted on bias in Europe means that little attention has been paid to it. There must be other stronger evidence to support this sentence.
Lines 396-397: Please provide at least a reference to support this sentence.
The study needs minimum editing. I have highlighted some areas in my comments.
Author Response
Response to Reviewer of Manuscript ijerph-2480502
Dear Reviewer
Thanks for your thorough reading and the useful comments to our article on unconscious bias. Below you find your comments and recommendations and our responses. We also submit a version of the manuscript in track changes mode.
-------------------------------------------------------------------------------------------------------------
Moderate editing of English language required
Response: Thanks for giving us hints where to edit the English language. We followed your recommendations.
Lines 80-81: The sentence here does not read well. Please reconstruct it.
Response: Sentence was reconstructed. It now reads “In healthcare biases and stereotypes may shape education when students learn with typical examples based on prevalence data, as Marcelin et al. point out.”
Lines 93-96: I commend the authors for mentioning the difference between their study and FitzGerald and Hurst.
Response: We mentioned how our study is different from that of FitzGerald and Hurst in lines 100 through 106. Furthermore, our study encompasses publications after March 2013, the date when FitzGerald and Hurst ended their search.
Lines 100-101: FitzGerald and Hurst did a systematic review. It would have been better if the authors did systematic review too in order to be able to compare the two studies appropriately. However, the scoping review is fine.
Response: Thanks for your recommendation. We are aware about the advantages of a systematic and a scoping review. Our scoping review is similar to the review by Fitzgerald and Hurst and is systematic in the sense that we follow a search protocol and gather all studies on the topic. We, however, decided to present our results in the form of a scoping review as we realize that many concepts are not well defined in the field of unconscious bias. We did not want to make the pretense that clear results can be gathered from the literature. To the contrary, we think that research is needed to clarify some of the concepts and tests used in order to clarify where the border between conscious and unconscious bias lies.
We also wanted to focus on the knowledge gaps to have a basis where future research needs to be undertaken and with what approaches unconscious bias can be tackled in the future.
Line 180: This is not clear. A prospective design is also a longitudinal design, so why n=1 and n=1 respectively?
Response: A prospective design is not necessarily similar to a longitudinal design. Prospective would entail to test a hypothesis at a future point in time. That does not mean that the study per se will be longitudinal, that is lasting longer.
Lines 209-210: What could have led to this difference in outcome?
Response: Thanks for this question. We discuss potential reasons for the observed difference in the results of the IAT and vignette studies in the discussion section. After summarizing previous findings on the differential predictability of implicit and explicit measures in lines 471-473 (“in situations which deliberate decisions are made explicit measures of attitudes and behavior intentions are more reliable than implicit measures which in turn are better to predict spontaneous reactions”), we come back to the seemingly contradictory results in lines 475-477 (“In the present review studies that used an IAT, and vignettes are coming in 1/3 of the cases to contradictory results. Both, deliberate as well as spontaneous decisions are common in the daily life of health professionals.”). In sum, we argue in our discussion that whereas the IAT is more predictive of behavior in situations that are driven by automatic processes (e.g., when strong habits are in place and a person is stressed, under time pressure, alcoholized, or tired), the vignettes should tap into unconscious processes that are ongoing in situations that are driven by more reflective information processes. We hope that this explanation provides at least a conceptual explanation for the observed differences in outcome.
Lines 239-240: The authors should re-write this sentence because it is not clear.
Response: The sentence was rewritten to make it clearer. It now reads “Six studies of these used both an implicit measure and vignettes while two were based on vignettes only and one on an implicit measure.”
Lines 321-324: This means that both the vignettes and IAT are not in the same direction. The question is, which of them is reliable?
Response: As there is no gold standard, we could not say that the vignettes or the IAT studies are more valid. This question will be part of a further analysis that we are currently preparing for publication as many issues could have an impact on the result that vignettes and IAT point to different directions (see response above). In the present discussion, we outline that whereas implicit measures have advantages with respect to the internal validity they at the same time have been criticized for their lack of external validity (lines 467-470: “These have the advantage of internal validity but have been criticized for their lack of external validity [117]. The IAT has shown validity in predicting bias (prejudice and stereotyping) more accurately than self-report [118]. However, there is a debate if these implicit measures can predict behavior.” Although this discussion is not conclusive, we hope that we can call attention to the reliability questions that you raised. We hope this answers the questions.
Lines 390-391: I am not sure that the fact that few studies were conducted on bias in Europe means that little attention has been paid to it. There must be other stronger evidence to support this sentence.
Response: Thanks for this argument. We agree that we cannot infer this statement from the mere number of studies. One could assume that unconscious bias is less a problem in Europe but there is no data so we could neither come to that conclusion. Yet, we can state that the topic was neglected in research in Europe. The sentence was rewritten accordingly. (“In contrast, the topic has been as widely examined by researchers in Europe or elsewhere in the world.”)
Lines 396-397: Please provide at least a reference to support this sentence.
Response: We agree that a reference should be provided for this sentence. However, as think that the heterogeneity does not need further (speculative) explanations, we deleted the sentences in lines 396-400 (“The subject of unconscious bias as well as discrimination have raised less attention in other parts of the world apart from the United States. Reasons might be that health disparities are less evident or less perceptible or that awareness and civic right movements are at different stages in other countries and that health disparities are not perceived as a problem, yet.”).
Comments on the Quality of English Language
The study needs minimum editing. I have highlighted some areas in my comments.
Response: Thanks for your comments. We complied with your suggestions and edited the few sentences you mentioned. Aside from that we can assure you that the paper was edited by a professional academic native speaker from the US who is active in research and publication.

Reviewer 3 Report
Thank you for the opportunity to read and review an interesting article on unconscious bias among health professionals. I have a few comments for the authors:
Abstract:
- Please reassess if it makes sense to write the numeruses.
- Please feel free to exclude individual sub-headings.
Results:
- I do not think that chapter 3.1 is necessary as you have already covered all the information in the Method.
- I am wondering if it might make sense to move the information presented in Section 3.2 into a table.
- Similarly, I am wondering if it would be useful to supplement Table 1 with the demographic data you provide above it.
- Overall, I think the data, while very informative, is rather fragmented and therefore less transparent. I am wondering if it might make sense to move Table 5 more towards the beginning of the results, so that the reader can get a quick overview of the studies, and then you report on the details.
Author Response
Responses or Reviewer of Manuscript ijerph-2480502
Dear Reviewer
Thank you for your thorough reading and the useful comments and suggestions to our article on unconscious bias. Below you find your comments and recommendations and our responses. We also submit a version of the manuscript in track changes mode.
---------------------------------------------------------------------------------------------------------
Comments and Suggestions for Authors
Thank you for the opportunity to read and review an interesting article on unconscious bias among health professionals. I have a few comments for the authors:
Abstract:
- Please reassess if it makes sense to write the numeruses.
Response: We are not sure what exactly you want us to do. We could delete the numbers of papers but we think that gives the readers an overall idea of the number of studies in the field and how they are distributed.
- Please feel free to exclude individual sub-headings.
Response: Thank you for this suggestion. We left the sub-headings unchanged as colleagues who read the abstract reassured us that they help them to swiftly grasp the gist of the review.
Results:
- I do not think that chapter 3.1 is necessary as you have already covered all the information in the Method.
Response: Thank you for this comment. We disagree: In the Methods section we provide the information how, when and where we did our search. In section 3.1 we present the result of the actual research and the concrete numbers. Therefore, we have to keep chapter 3.1 and clearly separate the methods and the results.
- I am wondering if it might make sense to move the information presented in Section 3.2 into a table.
Response: Thank you for this suggestion. We thought about the pros and cons of such a change and did in the end decide to keep the information presented in Section 3.2. in the written paragraph as putting it into a table would have resulted in a rather complicated table and our approach was to have a rather comprehensive Table 1 but all additional tables should be reduced in order to make it easy for readers to navigate through the manuscript and the respective tables. We hope this is OK.
- Similarly, I am wondering if it would be useful to supplement Table 1 with the demographic data you provide above it.
Response: Thank you for this suggestion. We are not exactly sure what you mean. Should we include the number of the participants age for all individual studies? This would enlarge the table and probably make it rather too comprehensive and maybe confusing. In order to keep the simplicity of the table (and thus make it easy for readers) we would like to stick with keeping the actual distribution of information in tables and written paragraphs.
- Overall, I think the data, while very informative, is rather fragmented and therefore less transparent. I am wondering if it might make sense to move Table 5 more towards the beginning of the results, so that the reader can get a quick overview of the studies, and then you report on the details.
Response: Thank you for this suggestion, we agree with you and moved table 5 upfront. It will now be shown on page 6, right after the study characteristics as Table 1. We also moved the respective text upfront to keep the order of the written description of the results and the table consistent (i.e., first description, then table).
